

# Clinical relevance assessment of animal preclinical research (RAA) tool: development and explanation

Kurinchi S. Gurusamy[1,2], David Moher[3,4], Marilena Loizidou[1], Irfan Ahmed[5], Marc T. Avey[3,4], Carly C. Barron[3,4,6], Brian Davidson[1], Miriam Dwek[7], Christian Gluud[8], Gavin Jell[1], Kiran Katakam[8], Joshua Montroy[9], Timothy D. McHugh[10], Nicola J. Osborne[11], Merel Ritskes-Hoitinga[12], Kees van Laarhoven[13], Jan Vollert[14,15] and Manoj Lalu[9]

[1] Research Department of Surgical Biotechnology, University College London, London, England, UK
[2] Surgery and Interventional Trials Unit, University College London, London, England, UK
[3] Centre for Journalology, Clinical Epidemiology Program, Ottawa Hospital Research Institute, The Ottawa Hospital, Ottawa, ON, Canada
[4] School of Epidemiology and Public Health, Faculty of Medicine, University of Ottawa, Ottawa, ON, Canada
[5] Department of Surgery, NHS Grampian, Aberdeen, Scotland, UK
[6] Department of Medicine, McMaster University, Hamilton, ON, Canada
[7] School of Life Sciences, University of Westminster, London, England, UK
[8] Copenhagen Trial Unit, Centre for Clinical Intervention Research, Rigshospitalet, Copenhagen University Hospital, Copehagen, Denmark
[9] Department of Anesthesiology and Pain Medicine, Blueprint Translational Research Group, Clinical Epidemiology and Regenerative Medicine Programs, Ottawa Hospital Research Institute, Ottawa Hospital, Department of Cellular and Molecular Medicine, University of Ottawa, Ottawa, ON, Canada
[10] UCL Centre for Clinical Microbiology, Division of Infection & Immunity, University College London, London, England, UK
[11] Responsible Research in Practice, London, England, UK
[12] SYRCLE, Department for Health Evidence, Radboud University Medical Center, Nijmegen, Netherlands
[13] Department of Surgery, Radboud Institute for Health Sciences, Nijmegen, Netherlands
[14] Pain Research, Department of Surgery & Cancer, Imperial College, London, England, UK
[15] Center of Biomedicine and Medical Technology Mannheim CBTM, Medical Faculty Mannheim, Heidelberg University, Mannheim, Germany

Corresponding author
Kurinchi S. Gurusamy,
k.gurusamy@ucl.ac.uk

## ABSTRACT

**Background:** Only a small proportion of preclinical research (research performed in animal models prior to clinical trials in humans) translates into clinical benefit in humans. Possible reasons for the lack of translation of the results observed in preclinical research into human clinical benefit include the design, conduct, and reporting of preclinical studies. There is currently no formal domain-based assessment of the clinical relevance of preclinical research. To address this issue, we have developed a tool for the assessment of the clinical relevance of preclinical studies, with the intention of assessing the likelihood that therapeutic preclinical findings can be translated into improvement in the management of human diseases.

**Methods:** We searched the EQUATOR network for guidelines that describe the design, conduct, and reporting of preclinical research. We searched the references of

these guidelines to identify further relevant publications and developed a set of domains and signalling questions. We then conducted a modified Delphi-consensus to refine and develop the tool. The Delphi panel members included specialists in evidence-based (preclinical) medicine specialists, methodologists, preclinical animal researchers, a veterinarian, and clinical researchers. A total of 20 Delphi-panel members completed the first round and 17 members from five countries completed all three rounds.

**Results:** This tool has eight domains (construct validity, external validity, risk of bias, experimental design and data analysis plan, reproducibility and replicability of methods and results in the same model, research integrity, and research transparency) and a total of 28 signalling questions and provides a framework for researchers, journal editors, grant funders, and regulatory authorities to assess the potential clinical relevance of preclinical animal research.

**Conclusion:** We have developed a tool to assess the clinical relevance of preclinical studies. This tool is currently being piloted.

## INTRODUCTION

Only a small proportion of preclinical research (research performed on animals prior to clinical trials) translates into clinical benefit in humans. In a study evaluating the translation of preclinical research into clinical benefit, a total of 101 technologies (including drugs, devices, and gene therapy) were assessed in preclinical models and considered to be promising. Of these, 27 (27%) were subsequently tested in human randomised clinical trials within 20 years of their preclinical publication. Of these 27 human translational attempts, only one technology resulted in clinical benefit (1%; 95% confidence interval [0.2–5.4]) (*Contopoulos-Ioannidis, Ntzani & Ioannidis, 2003*).

In a 2014 report, of 100 potential drugs giving objective improvement when evaluated in a commonly used mouse model for the treatment of amyotrophic lateral sclerosis, none were found to be clinically beneficial (*Perrin, 2014*). Finally, a systematic review found that there were significant differences in the estimates of treatment effectiveness in animal experiments compared to that observed in human randomised controlled trials with some interventions being beneficial in animals but harmful in humans (*Perel et al., 2007*). Some of the reasons for the lack of translation of the beneficial results observed in preclinical research into human clinical benefit could relate to the design, conduct, and reporting of preclinical studies (*Collins & Tabak, 2014*; *Begley & Ioannidis, 2015*; *Ioannidis, 2017*). Further information of the reasons and explanations for the lack of translation of the beneficial results observed in preclinical research into human clinical benefit is provided under the explanations for the relevant domains and signalling questions.

### Why is this project needed?

A domain-based tool is a tool that assesses different aspects that impact the outcome of interest (in this case, clinical relevance of preclinical research). Such domain-based tools are preferred by methodologists to assess clinical studies (*Higgins & Green, 2011*; *Whiting et al., 2011*, *2016*; *Sterne et al., 2016*); however, as indicated below, no such tool exists to assess the potential clinical relevance of preclinical research.

### Aim of this project

The aim of this project was to design a domain-based tool to assess the clinical relevance of a preclinical research study in terms of the likelihood that therapeutic preclinical findings can be translated into improvement in the management of human diseases. As part of the process, the scope and applicability of this tool was defined to include only in vivo animal interventional studies.

### Who is this intended for?

This tool is intended for all preclinical researchers and clinical researchers considering translation of preclinical findings to first-in-human clinical trials, the funders of such studies, and regulatory agencies that approve first-in-human studies.

## MATERIALS AND METHODS

We followed the Guidance for Developers of Health Research Reporting Guidelines (*Moher et al., 2010*) as there is no specific guidance for developers of tools to assess clinical relevance of preclinical tools. The registered protocol is available at http://doi.org/10.5281/zenodo.1117636 (Zenodo registration: 1117636). The study did not start until the protocol for the current study was registered. The overall process is summarised in Fig. 1.

### Search methods

First, we established whether there is any domain-based assessment tool for preclinical research. We searched the EQUATOR Network's library of reporting guidelines using the terms 'animal' or 'preclinical' or 'pre-clinical'. We included any guidelines or tools that described the design, conduct, and reporting of preclinical research. We searched the references of these guidelines to identify further relevant publications. We searched only the EQUATOR Network's library as it contains a comprehensive search of the existing reporting guidelines. A scoping search of Pubmed using the terms 'animal[tiab] AND (design[tiab] OR conduct[tiab] OR report[tiab])' returned nearly 50,000 records and initial searching of the first 1,000 of them did not indicate any relevant publications. Therefore, the more efficient strategy of searching the EQUATOR Network's library was used to find any publications of a domain-based tool related to design, conduct, or reporting guidelines of preclinical research.

### Development of domains and signalling questions

We recorded the topics covered in the previous guidance on preclinical research to develop a list of domains and signalling questions to be included in the formal domain-based
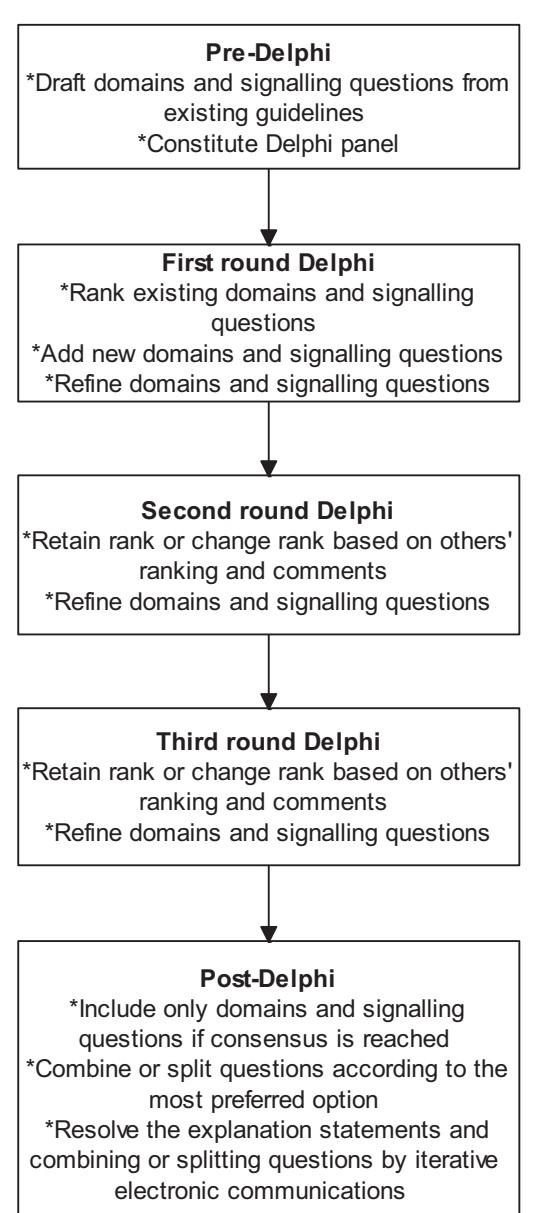

**Figure 1 Overall process.** The outline of the process is shown in this figure. A total of three rounds were conducted. Consensus agreement was reached when at least 70% of panel members strongly agreed (scores of 7 or more) to include the domain or signalling question.

assessment of preclinical research. The first author identified and included all the topics covered in each of the publications and combined similar concepts. The initial signalling questions were developed after preliminary discussions with and comments from the all the Delphi panel members (please see below) prior to finalising the initial list of signalling questions. The full list of the topics covered in the publications and the initial signalling questions are available in the Supplemental Information Appendix 1 (second column). The signalling questions are questions that help in the assessment of a

domain. Additional details about how domains and signalling questions can be used are listed in Box 1.

**Box 1**

1. Signalling questions are questions that help in the assessment about a domain. As such, the overall domain assessment is more important that the answers for individual signalling questions.

2. Depending upon the nature and purpose for the research, certain domains may be more important than the other. For example, if the purpose is to find out whether there is enough information to perform a first-in-human study, the clinical translatability and reproducibility domain is of greater importance than if the report was about the first interventional study on a newly developed experimental model.

## Selection of experts and consensus

Next, we approached experts in the field of preclinical and clinical research to participate in the development process. The group of experts were purposively sampled using snowballing principles (used to identify people with a rich knowledge base on a topic) (*Heckathorn, 2011*): people who perform only preclinical research, people who perform only clinical research, people who perform both preclinical and clinical research, and methodologists, all of whom had interest in improving the clinical relevance of preclinical research were approached and asked to suggest other experts who could contribute to the process. We conducted a modified Delphi-consensus method to refine and develop the tool. The Delphi-consensus method was based on that described by *Jones & Hunter (1995)*. The steps in the Delphi process is shown in box 2. All were completed electronically using an excel file.

**Box 2**

1. The first round included questions regarding scope and necessity (i.e. should the tool include all types of preclinical research or only preclinical in vivo animal research and whether a domain or signalling question should be included in the final tool) in addition to the signalling questions available in the second column of Appendix 1.

2. The signalling questions were already classified into domains and were supported by explanations and examples in the first Delphi round. The original classification of signalling questions is available in Appendix 1.

3. The Delphi panel ranked the questions by importance on a scale of 1–9 with 1 being of lowest importance and 9 being highest importance.

4. The ranking scores were then grouped into three categories: 1–3 being strong disagreement about the importance of the question, 4–6 being weak to moderate agreement about the question, and 7–9 being strong agreement about the question. The questions were phrased in such a way that higher scores supported inclusion into the tool and lower scores indicated exclusion (of the scope, domain, or signalling

question). Consensus was considered to have been reached when 70% or more participants scored 7 or more. There is variability in the definition of consensus and 70% or more participants scoring 7 or more is within the previously reported range for consensus agreement (*Sumsion, 1998*; *Hasson, Keeney & McKenna, 2000*; *Diamond et al., 2014*). This is a commonly used percentage for defining consensus in the Delphi process (*Kleynen et al., 2014*; *Kirkham et al., 2017*).

5. A total of three rounds were conducted. The panel members were allowed to add new domains or signalling questions in the first round. The panel members could also suggest revisions to the existing domains or questions (for example revision of the explanation, examples, or by combining some domains or questions and splitting others) in all the rounds.

6. After the first round, the Delphi panel were shown their previous rank for the question and the median rank (and interquartile range) of questions of the Delphi panel.
In addition, the Delphi panel were also asked to choose the best version of any revisions to the questions and provide ranks for any additional questions identified in the first round.

7. The panel members were able to retain or change the rank in each of the rounds after the first round.

8. For calculation of median and interquartile range of ranks and consensus, non-responses were ignored.

9. At the end of the third round, the aspects which had been ranked with a score of 7 or above for necessity by at least 70% of the panel were included in the final tool.

10. There was no restriction on the Delphi panel to consult others while ranking the questions. However, only one final response on the set of questions was accepted from each Delphi panel member.

Then, we refined the signalling questions and explanation by iterative electronic communications. Finally, we piloted the tool in biomedical researchers who perform animal preclinical research and those who perform first-in-human studies to clarify the signalling questions and explanations.

## RESULTS

### Deviations from protocol

There were two deviations from our protocol. Firstly, we did not exclude questions even when consensus was reached on the necessity of the questions: this was because the phrasing of the domain/signalling question, the explanation, the domain under which the signalling question is located, and combining or splitting the domains were still being debated. Secondly, we did not conduct an online meeting of the panel members between the second and third rounds of the Delphi process because listing and summarising the comments from different contributors achieved the aim of providing information to justify or revise the ranking.

## Search results

Twenty-one publications were identified (*Idris et al., 1996*; *Sena et al., 2007*; *Bath, Macleod & Green, 2009*; *Fisher et al., 2009*; *Macleod et al., 2009*; *Bouxsein et al., 2010*; *Hooijmans, Leenaars & Ritskes-Hoitinga, 2010*; *Kilkenny et al., 2010*; *Van der Worp et al., 2010*; *Begley & Ellis, 2012*; *Landis et al., 2012*; *Hooijmans et al., 2014*; *National Institutes of Health, 2014*; *Perrin, 2014*; *Bramhall et al., 2015*; *Czigany et al., 2015*; *Andrews et al., 2016*; *Biophysical Journal, 2017*; *Open Science Framework, 2017*; *Osborne et al., 2018*; *Smith et al., 2018*). The main topics covered in these publications were bias, random errors, reproducibility, reporting, or a mixture of these elements which result in lack of translation of preclinical research into clinical benefit. One publication was based on a consensus meeting (*Andrews et al., 2016*) and five were based on expert working groups (*Hooijmans, Leenaars & Ritskes-Hoitinga, 2010*; *Kilkenny et al., 2010*; *Landis et al., 2012*; *National Institutes of Health, 2014*; *Osborne et al., 2018*); and the remaining were opinions of the authors (*Idris et al., 1996*; *Sena et al., 2007*; *Bath, Macleod & Green, 2009*; *Fisher et al., 2009*; *Macleod et al., 2009*; *Bouxsein et al., 2010*; *Van der Worp et al., 2010*; *Begley & Ellis, 2012*; *Hooijmans et al., 2014*; *Perrin, 2014*; *Bramhall et al., 2015*; *Czigany et al., 2015*; *Biophysical Journal, 2017*; *Open Science Framework, 2017*; *Smith et al., 2018*). All five publications based on consensus meeting or expert working groups were reporting guidelines (*Hooijmans, Leenaars & Ritskes-Hoitinga, 2010*; *Kilkenny et al., 2010*; *Landis et al., 2012*; *National Institutes of Health, 2014*; *Osborne et al., 2018*).

## Survey respondents

A total of 20 Delphi-panel members completed the first round and 17 members from 5 countries completed all three rounds. The panel members included specialists representing a broad scope of stakeholders, including target users that would evaluate interventions for potential 'bench-to-bedside' translation: evidence-based (preclinical) medicine specialists, methodologists, preclinical researchers, veterinarian, and clinical researchers from UK, Canada, Denmark, and Netherlands. The mean and standard deviation age of people who completed was 48.4 and 10.9 at the time of registration of protocol. Of the 17 respondents completing all the three rounds, 12 were males and 5 were females; eleven of these 17 respondents were Professors or had equivalent senior academic grade at the time of registration. There were no conflicts of interest for the survey respondents other than those listed in the "Conflicts of Interest" section of this document.

The reasons for drop-out included illness (one member) and concerns about the scope and applicability of the tool (two members). These were aspects that were developed as part of the process of the registered protocol. Therefore, clarity on the scope and applicability was available only at the end of the Delphi process and not in the first round of the Delphi-process.

## Domains and signalling questions

The Delphi panel agreed on eight domains, which constitutes the tool. Table 1 lists the domains and signalling questions for which consensus agreement was reached. The first

**Table 1 Domains and signalling questions.**

| Domain or signalling question | Classification |
|---|---|
| **Domain 1: Clinical translatability of results to human disease or condition (construct validity)** | **Low concern/Moderate concern/High concern** |
| 1.1 Did the authors use a model that adequately represents the human disease? | 'Yes'/'Probably yes'/'Probably no'/'No'/'No information' |
| 1.2 Did the authors sufficiently identify and characterise the model? | 'Yes'/'Probably yes'/'Probably no'/'No'/'No information' |
| 1.3 Were the method and timing of the intervention in the specific model relevant to humans? | 'Yes'/'Probably yes'/'Probably no'/'No'/'No information' |
| 1.4 If the study used a surrogate outcome, was there a clear and reproducible relationship between an intervention effect on the surrogate outcome (measured at the time relationship chosen in the preclinical research) and that on the clinical outcome? | 'Not applicable'/'Yes'/'Probably yes'/'Probably no'/'No'/'No information' |
| 1.5 If the study used a surrogate outcome, did previous experimental studies consistently demonstrate that change in surrogate outcome(s) by a treatment led to a comparable change in clinical outcomes? | 'Not applicable'/'Yes'/'Probably yes'/'Probably no'/'No'/'No information' |
| 1.6 Did a systematic review with or without meta-analysis demonstrate that the effect of an intervention or a similar intervention on a preclinical model was similar to that in humans? | 'Yes'/'Probably yes'/'Probably no'/'No'/'No information' |
| **Domain 2: Experimental design and analysis** | **Low concern/Moderate concern/High concern** |
| 2.1 Did the authors describe sample size calculations? | 'Yes'/'Probably yes'/'Probably no'/'No'/'No information' |
| 2.2 Did the authors plan and perform statistical tests taking the type of data, the distribution of data, and the number of groups into account? | 'Yes'/'Probably yes'/'Probably no'/'No'/'No information' |
| 2.3 Did the authors make adjustment for multiple hypothesis testing? | 'Yes'/'Probably yes'/'Probably no'/'No'/'No information' |
| 2.4 If a dose-response analysis was conducted, did the authors describe the results? | 'Not applicable'/'Yes'/'Probably yes'/'Probably no'/'No'/'No information' |
| 2.5 Did the authors assess and report accuracy? | 'Yes'/'Probably yes'/'Probably no'/'No'/'No information' |
| 2.6 Did the authors assess and report precision? | 'Yes'/'Probably yes'/'Probably no'/'No'/'No information' |
| 2.7 Did the authors assess and report sampling error? | 'Yes'/'Probably yes'/'Probably no'/'No'/'No information' |
| 2.8 Was the measurement error low or was the measurement error adjusted in statistical analysis? | 'Yes'/'Probably yes'/'Probably no'/'No'/'No information' |
| **Domain 3: Bias (internal validity)** | **Low concern/Moderate concern/High concern** |
| 3.1 Did the authors minimise the risks of bias such as selection bias, confounding bias, performance bias, detection bias, attrition bias, and selective outcome reporting bias? | 'Yes'/'Probably yes'/'Probably no'/'No'/'No information' |
| **Domain 4: Reproducibility of results in a range of clinically relevant conditions (external validity)** | **Low concern/Moderate concern/High concern** |
| 4.1 Were the results reproduced with alternative preclinical models of the disease/condition being investigated? | 'Yes'/'Probably yes'/'Probably no'/'No'/'No information' |
| 4.2 Were the results consistent across a range of clinically relevant variations in the model? | 'Yes'/'Probably yes'/'Probably no'/'No'/'No information' |
| 4.3 Did the authors report take existing evidence into account when choosing the comparators? | 'Yes'/'Probably yes'/'Probably no'/'No'/'No information' |
| **Domain 5: Reproducibility and replicability of methods and results in the same model** | **Low concern/Moderate concern/High concern** |
| 5.1 Did the authors describe the experimental protocols/methods sufficiently to allow their replication? | 'Yes'/'Probably yes'/'Probably no'/'No'/'No information' |

| Domain or signalling question | Classification |
|---|---|
| 5.2 Did an independent group of researchers replicate the experimental protocols/methods? | 'Yes'/'Probably yes'/'Probably no'/'No'/'No information' |
| 5.3 Did the authors or an independent group of researchers reproduce the results in similar and different laboratory conditions? | 'Yes'/'Probably yes'/'Probably no'/'No'/'No information' |
| **Domain 6: Implications of the study findings (study conclusions)** | **Low concern/Moderate concern/High concern** |
| 6.1 Did the authors' conclusions represent the study findings, taking its limitations into account? | 'Yes'/'Probably yes'/'Probably no'/'No'/'No information' |
| 6.2 Did the authors provide details on additional research required to conduct first-in-human studies? | 'Yes'/'Probably yes'/'Probably no'/'No'/'No information' |
| **Domain 7: Research integrity** | **Low concern/Moderate concern/High concern** |
| 7.1 Did the research team obtain ethical approvals and any other regulatory approvals required to perform the research prior to the start of the study? | 'Yes'/'Probably yes'/'Probably no'/'No'/'No information' |
| 7.2 Did the authors take steps to prevent unintentional changes to data? | 'Yes'/'Probably yes'/'Probably no'/'No'/'No information' |
| **Domain 8: Research transparency** | **Low concern/Moderate concern/High concern** |
| 8.1 Did the authors describe the experimental procedures sufficiently in a protocol that was registered prior to the start of the research? | 'Yes'/'Probably yes'/'Probably no'/'No'/'No information' |
| 8.2 Did the authors describe any deviations from the registered protocol? | 'Yes'/'Probably yes'/'Probably no'/'No'/'No information' |
| 8.3 Did the authors provide the individual subject data along with explanation for any numerical codes/substitutions or abbreviations used in the data to allow other groups of researchers to analyse? | 'Yes'/'Probably yes'/'Probably no'/'No'/'No information' |

four domains relate to the study design and analysis that are within the control of the research team (clinical translatability of results to human disease or condition (construct validity)), experimental design and data analysis, bias (internal validity), and reproducibility of results in a different disease-specific model (external validity). The fifth domain relates to replicability of results for which the research team may have to rely on other research teams (reproducibility and replicability of methods and results in the same model); however, these aspects can be integrated as part of the same study. The sixth domain relates to study conclusions which considers the study design, analysis, and reproducibility and replicability of results. The last two domains relate to factors that increase or decrease the confidence in the study findings (research integrity and research transparency).

These eight domains cover a total of 28 signalling questions. The number of questions in each domain range from 1 to 8, with a median of three questions in each domain. All the signalling questions have been phrased in such a way that a classification of 'yes' or 'probably yes' will result in low concerns about the clinical relevance of the study for the domain.

## Scope and applicability of the tool

The scope of the tool is only for assessment of the clinical relevance of a preclinical research study in terms of the likelihood that therapeutic preclinical findings can be

translated into improvement in the management of human diseases and not for assessment of the quality of the study, that is how well the study was conducted, although we refer to tools that assess how well the study was conducted. It is important to make this distinction as even a very well-designed and conducted preclinical study may not translate to improvement in the management of human diseases, as is the case of clinical research.

As part of the Delphi process, the scope was narrowed to include only in vivo laboratory based preclinical animal research evaluating interventions. Therefore, our tool is not intended for use on other forms of preclinical research such as in vitro work (e.g. cell cultures), in silico research, or veterinary research. This tool is not applicable in the initial exploratory phase of development of new animal models of disease, although the tool is applicable in interventional studies using such newly developed models.

The domains and signalling questions in each round of the Delphi process and post-Delphi process are summarised in Fig. 2.

### Classification of signalling questions and domains

Consistent with existing domain based tools, responses to each signalling question can be classified as 'yes', 'probably yes', 'probably no', 'no', or 'no information' (Sterne et al., 2016; Whiting et al., 2016), depending upon the information described in the report or after obtaining the relevant information from the report's corresponding author, although the study authors may provide answers that the assessor asks because of cognitive bias. A few questions can also be classified as 'not applicable'. These questions start with the phrase 'if'. For classification of the concerns in the domain, such questions are excluded from the analysis.

A domain can be classified as 'low concern' if *all* the signalling questions under the domain were classified as 'yes' or 'probably yes', 'high concern' if *any* of the signalling questions under the domain were classified as 'no' or 'probably no', and as 'moderate concern' for all other combinations.

### Overall classification of the clinical relevance of the study

A study with 'low concerns' for all domains will be considered as a study with high clinical relevance in terms of translation of preclinical results with similar magnitude and direction of effect to improve management of human diseases. A study with unclear or high concerns for one or more domains will be considered as a study with uncertain clinical relevance in terms of translation of preclinical results with similar magnitude and direction of effect to improve management of human diseases.

However, depending upon the nature and purpose for use of the research, certain domains may be more important than the others, and the users can decide in advance whether a particular domain is important (or not). For example if the purpose is to find out whether there is enough information to perform a first-in-human study, the clinical translatability and reproducibility domain is of greater importance than if the report was about the first interventional study on the model.

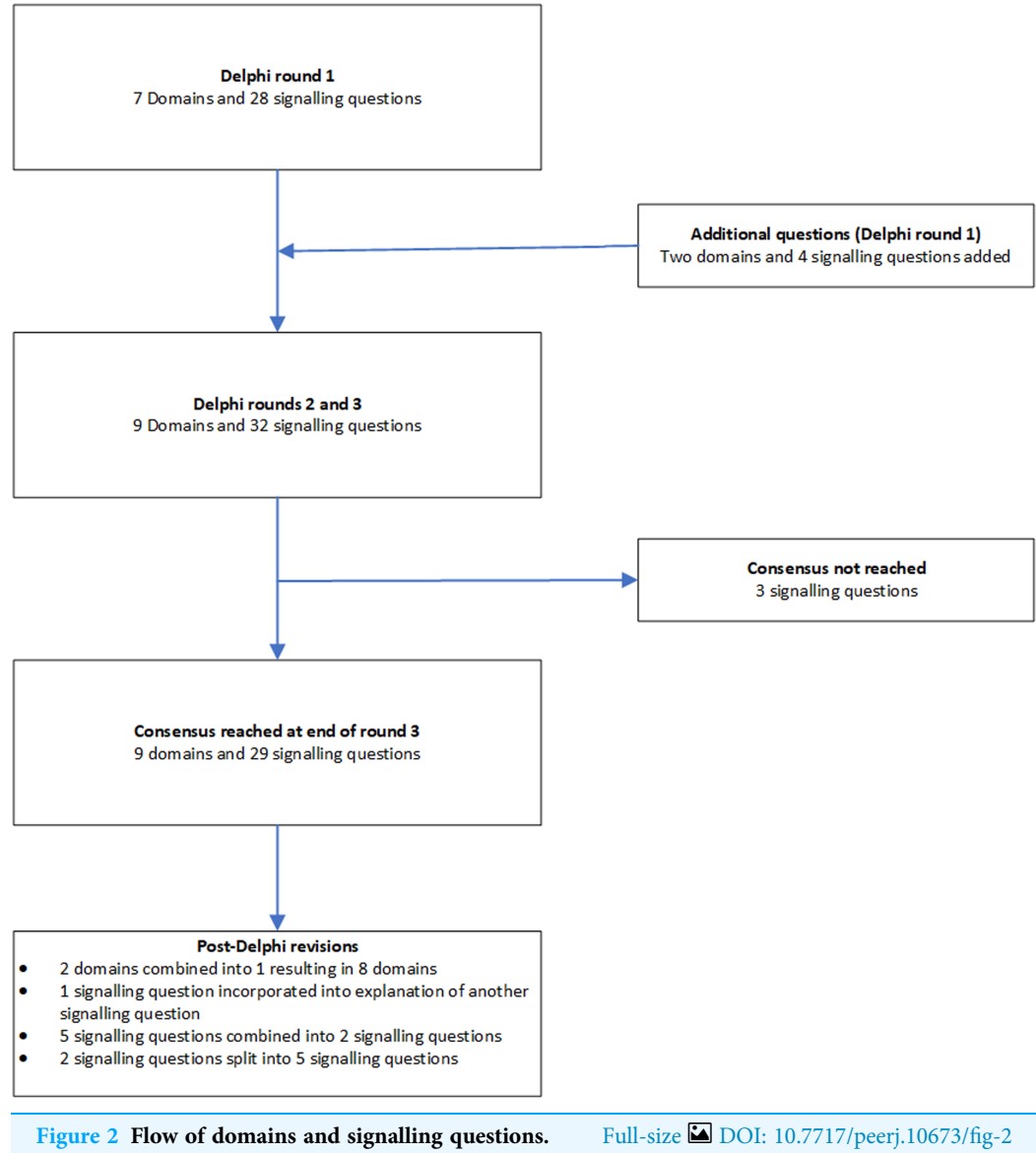

**Figure 2 Flow of domains and signalling questions.**

At the design and conduct stage, researchers, funders, and other stakeholders can specifically look at the domains that are assessed as unclear or high concern and improve the design and conduct to increase the clinical relevance. At the reporting stage, researchers, funders, and other stakeholders can use this tool to design, fund, or give approval for further research.

## Practical use of the tool

The tool should be used with a clinical question in mind. This should include the following aspects of the planned clinical study as a minimum: population in whom the intervention or diagnostic test is used, intervention and control, and the outcomes (PICO).

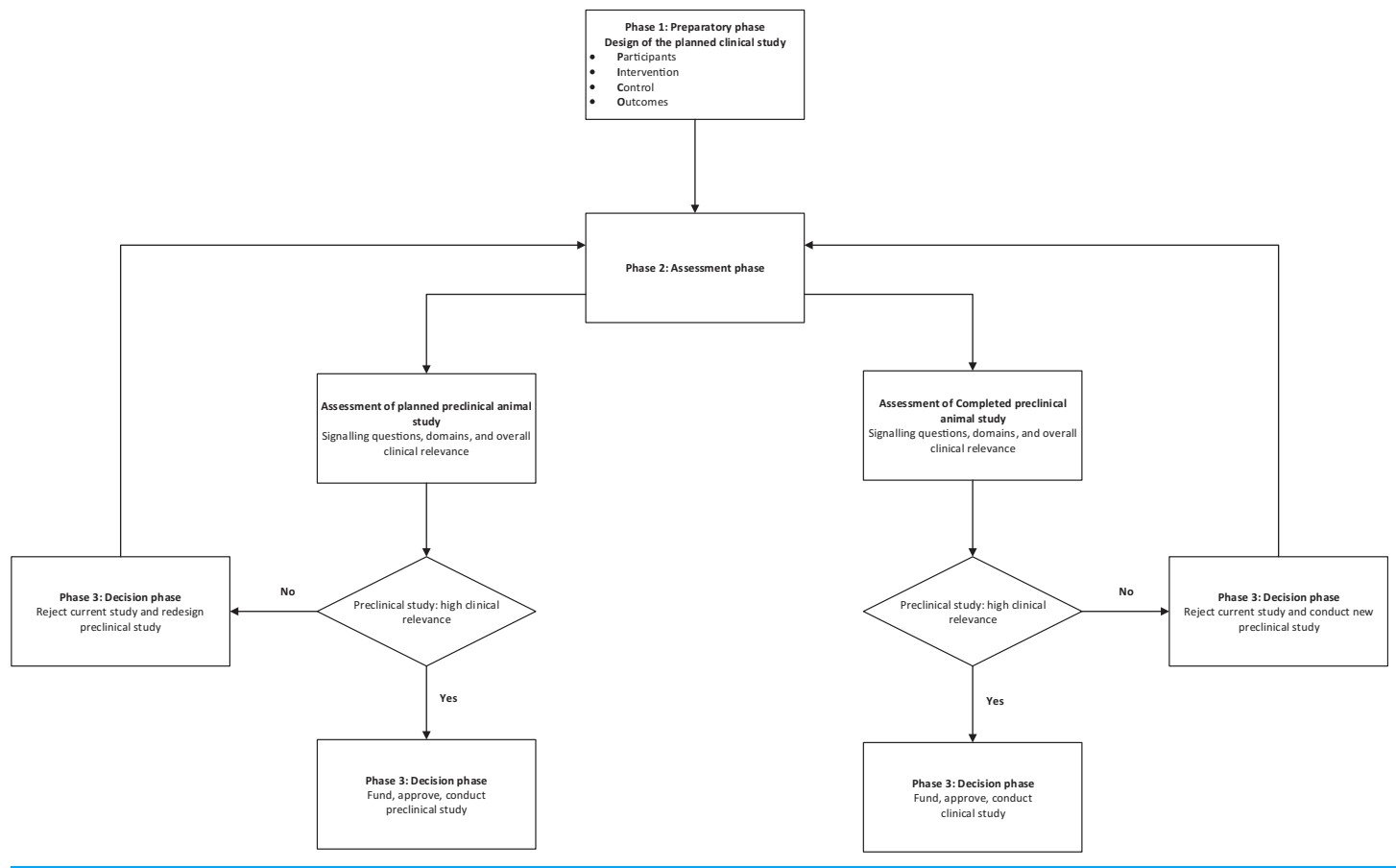

**Figure 3 Schema for use of the tool.**

We recommend that the tool is used after successfully completing the training material, which includes examples of how the signalling questions can be answered and assessment of understanding the use of the tool (the training material is available at: https://doi.org/10.5281/zenodo.4159278) and at least two assessors using the tool independently.

A schema for the practical use of the tool is described in Fig. 3.

### *Scoring*

The tool has not been developed to obtain an overall score for clinical relevance assessment. Therefore, modifying the tool by assigning scores to individual signalling questions or domains is likely to be misleading.

### Panel agreement

Appendix 1 summarises the Delphi panel agreement on the different domains and signalling questions. As shown in the Appendix 1, the domains, the signalling questions, and the terminologies used have improved significantly from the starting version of the tool. Appendix 1 also demonstrates that there was a change in the agreement in the questions indicating that the panel members were receptive to others' views while ranking the questions.

# RATIONALE AND EXPLANATION OF DOMAINS AND SIGNALLING QUESTIONS

## Domain 1: clinical translatability of results to human disease or condition (construct validity)

The purpose of this domain is to assess whether statistically positive results in the reports of the preclinical animal research studies could result in clinical benefit. This evaluation focuses on both primary outcomes and secondary outcomes, or the 'main findings' if the reports do not explicitly declare primary and secondary outcomes.

### 1.1 Did the authors use a model that adequately represents the human disease?

This question assesses biological plausibility. We have used the term 'model' to refer to the animal model used as a substitute for human disease, for example a mouse model of multiple myeloma. We have also used this term to refer to induction methods in animals in a non-diseased state that progress to a diseased state, for example a rat model of behavioural alterations (forced swim test) mimicking depression (*Yankelevitch-Yahav et al., 2015*), or animals which have been exposed to a treatment even if they did not have any induced human disease, for example a rabbit model of liver resection, a canine model of kidney transplantation. Studies have shown that animal researchers frequently use disease models that do not adequately represent or relate to the human disease (*De Vries et al., 2012*; *Sloff et al., 2014a*, *2014b*; *Zeeff et al., 2016*).

Specific characteristics to consider include species and/or strain used, age, immune competence, and genetic composition as relevant. Other considerations include different methods of disease induction in the same or different species.

This signalling question considers whether the researchers reporting the results ('authors') have described on information such as characteristics of model, different methods of disease induction (if appropriate), and biological plausibility while choosing the model, and have the researchers provided evidence for the choice of animal model. The assessment of these questions may require subject content expertise.

### 1.2 Did the authors identify and characterise the model?

This question assesses whether after choosing the appropriate model (species, sex, genetic composition, age), the authors have performed studies to characterise the model. For example sepsis is often induced through caecal ligation and puncture; however, the effects of this procedure can produce variable sepsis severity. Another example is when genes that induce disease may not be inherited reliably: the resulting disease manifestation could be variable and interventions may appear to be less effective or more effective than they actually are (*Perrin, 2014*). Therefore, it is important to ensure that the genes that induce the disease are correctly identified and that such genes are inherited. Another example is when the authors want to use a knockout model to understand the mechanism of how an intervention works based on the assumption that the only difference between the knockout mice and the non-knockout mice is the knockout gene.

However, the animals used may still contain the gene that was intended to be removed or the animals may have other genes introduced during the process of creating the knockout mice (*Eisener-Dorman, Lawrence & Bolivar, 2009*). Therefore, it is important to understand and characterise the baseline model prior to testing an experimental intervention.

### 1.3 Were the method and timing of the intervention in the specific model relevant to humans?

For pharmacological or biological interventions, this question refers to the dose and route of administration. For other types of interventions, such as surgery or device implementation, the question refers to whether the method used in the animal model is similar to that in humans.

For pharmacological interventions, there may be a therapeutic dose and route which is likely to be safe and effective in humans. It is unlikely that the exact dose used in animals is studied in humans, at least in the initial human safety studies. Therefore, dose conversion is used in first-in-human studies. Simple practice guides and general guidance for dose conversion between animals and humans are available (*United States Food and Drug Administration, 2005*; *Nair & Jacob, 2016*; *European Medicines Agency (EMA), 2017*). However, some researchers may use doses in animals at levels that would be toxic when extrapolated to humans and therefore unlikely to be used. Dose conversion guides (*Nair & Jacob, 2016*) can help with the assessment of whether the dose used is likely to be toxic. The effectiveness of an intervention at such toxic doses is not relevant to humans. It is preferable to use the same route of administration for animal studies as planned in humans, since different routes may lead to different metabolic fate and toxicity of the drug.

For non-pharmacological interventions for which similar interventions have not been tested in humans, feasibility of use in humans should be considered. For example thermal ablation is one of the treatment options for brain tumours. Ablation can, for example also be achieved by irreversible electroporation, which involves passing high voltage electricity and has been attempted in human liver and pancreas (*Ansari et al., 2017*; *Lyu et al., 2017*). However, the zone affected by irreversible electroporation has not been characterised fully: treatment of human brain tumours using this technique can only be attempted when human studies confirm that there are no residual effects of high voltage electricity in the surrounding tissue (not requiring ablation). Until then, the testing of irreversible electroporation in animal models of brain tumours is unlikely to progress to human trials and will not be relevant to humans regardless of how effective it may be.

The intervention may also be effective only at a certain time point in the disease (i.e. 'therapeutic window'). It may not be possible to recognise and initiate treatment during the therapeutic window because of the delays in appearance of symptoms and diagnosis. Therefore, there is no rationale in performing preclinical animal studies in which the intervention cannot be initiated during the likely therapeutic window. Finally, the treatment may be initiated prior to induction of disease in animal models: this may not reflect the actual use of the drug in the human clinical situation.

### 1.4 If the study used a surrogate outcome, was there a clear and reproducible relationship between an intervention effect on the surrogate outcome (measured at the time chosen in the preclinical research) and that on the clinical outcome?

A 'surrogate outcome' is an outcome that is used as a substitute for another (more direct) outcome along the disease pathway. For example in the clinical scenario, an improvement in CD4 count (surrogate outcome) leads to a decrease in mortality (clinical outcome) in people with human immune deficiency (HIV) (*Bucher et al., 1999*). The relationship between the effect of the intervention (a drug that improves the CD4 count) on the surrogate outcome (CD4 count) and a clinical outcome (mortality after HIV infection) should be high, should be shown in multiple studies, and should be independent of the type of intervention for a surrogate outcome to be valid (*Bucher et al., 1999*). This probably applies to preclinical research as well. For example the relationship between the effect of an intervention (a cancer drug) on the surrogate outcome (apoptosis) and a clinical outcome or its animal equivalent (for example mortality in the animal model) should be high, shown in multiple studies and independent of the type of intervention for a surrogate outcome to be valid in the preclinical model.

If the surrogate outcome is the only pathway or the main pathway between the disease, intervention, and the clinical outcome (or its animal equivalent) (Fig. 4), the surrogate outcome is likely to be a valid indirect surrogate outcome (*Fleming & DeMets, 1996*). This, however, should be verified in clinical studies. For example preclinical animal research studies may use gene or protein levels to determine whether an intervention is effective. If the gene (or protein) lies in the only pathway between the disease and animal equivalent of the clinical outcome, a change in expression, levels, or activity of the gene (or protein) is likely to result in an equivalent change in the animal equivalent of the clinical outcomes. To simplify this even further this signalling question can be simplified to the context in which it is used for example 'Is apoptosis at 24 h (surrogate outcome) in the preclinical animal model correlated with improved survival in animals (animal equivalent of a clinical outcome)'? Another example of this signalling question simplified to the context of the research can be 'Are aberrant crypt foci (surrogate outcome) in animal models correlated to colon cancer in these models (animal equivalent of a clinical outcome)'?

This signalling question assesses whether the authors have provided evidence for the relationship between surrogate outcome and the clinical outcome (or its animal equivalent). There is currently no guidance as to what a high level of association is in terms of determining the relationship between surrogate outcomes and the clinical outcomes (or its animal equivalent). Some suggestions are mentioned in Appendix 2.

### 1.5 If the study used a surrogate outcome, did previous experimental studies consistently demonstrate that change in surrogate outcome(s) by a treatment led to a comparable change in clinical outcomes?

This question aims to go further than the evaluation of association between surrogate outcome and the clinical outcome (or its animal equivalent). A simple association between

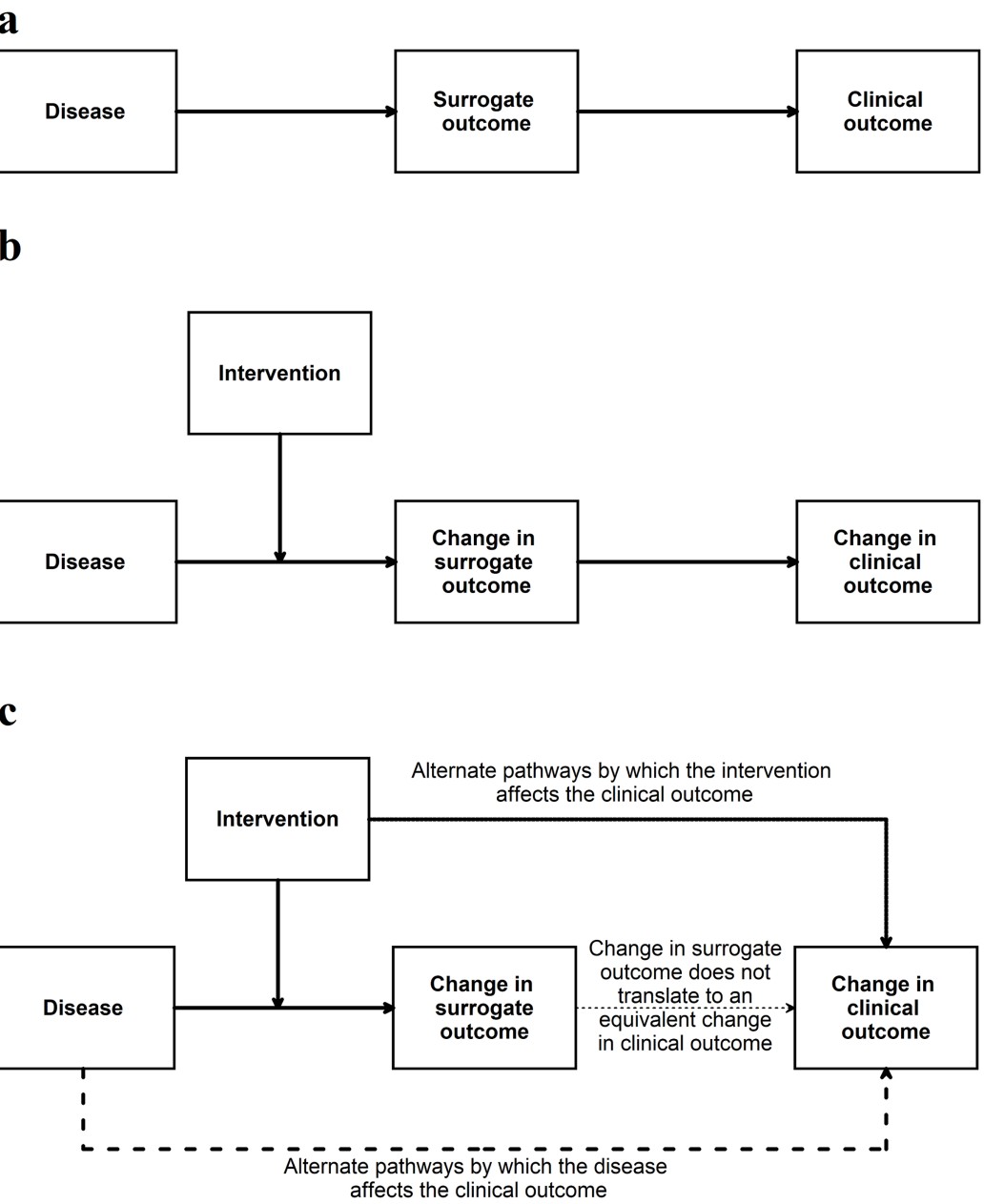

**Figure 4 Situation when a surrogate outcome is likely to be valid.** (A) The surrogate outcome is the only pathway that the disease can cause the clinical outcome. (B) The intervention acts in this pathway and causes a change in surrogate outcome, leading to a change in the clinical outcome. (C) If there are other pathways (which are not affected by the intervention) through which the disease can cause the clinical outcome, then the validity of the surrogate outcome will be decreased. If the intervention affects the clinical outcome through pathways unrelated to the surrogate outcome, then the validity of the surrogate outcome will be decreased.

a surrogate outcome and clinical outcome may be because the surrogate outcome may merely be a good predictor. For example sodium fluoride caused more fractures despite increasing bone mineral density, even though, low bone mineral density is associated with increased fractures (*Bucher et al., 1999*). If a change in the surrogate outcome by a

treatment results in a comparable change in the clinical outcome (or its animal equivalent), the surrogate outcome is likely to be a valid surrogate outcome (Fig. 4). This change has to be consistent, that is, most studies showing that a treatment results in a comparable improvement in the clinical outcome (or its animal equivalent). Note that it is possible that there may not a fully comparable change, for example a 50% improvement in the surrogate outcome may result only in a 25% improvement in the animal equivalent of the clinical outcome. In such situations, it is possible to use the 'proportion explained' approach proposed by *Freedman, Graubard & Schatzkin (1992)*, a concept which was extended to randomised controlled trials and systematic reviews by *Buyse et al. (2000)*. This involves calculating the association between the effect estimates of the surrogate outcome and clinical outcome (or its animal equivalent) from the different trials or centres within a trial (*Buyse et al., 2000*) (although, one can obtain a more reliable estimate of this association using individual participant data) (*Tierney et al., 2015*).

Generally, few surrogate outcomes are validated substitutes for clinical outcomes: an example of a valid surrogate outcome is CD4 count in people with human immune deficiency (HIV) (*Bucher et al., 1999*). Even if an association exists between the surrogate outcome and the clinical outcome, failure to demonstrate that changes in surrogate outcome by a treatment led to changes in clinical outcome can have disastrous effects (*Bucher et al., 1999*; *Yudkin, Lipska & Montori, 2011*; *Kim & Prasad, 2015*; *Rupp & Zuckerman, 2017*) (Appendix 3).

### 1.6 Did a systematic review with or without meta-analysis demonstrate that the effect of an intervention or a similar intervention in animal model was similar to that in humans?

The best way to find consistent evidence to support or refute the validity of surrogate outcomes (covered in the previous signalling questions) and the comparability of the animal equivalent of the clinical outcomes to that in humans is by systematic reviews. For example if an intervention results in better functional recovery in a mouse model of stroke, then does it also result in better functional recovery in humans with stroke? If so, other interventions can be tested in this model. Systematic reviews help in calculating the association between the effect estimates of the surrogate outcome and clinical outcome (or its animal equivalent) from the different trials or centres within a trial, as mentioned previously (*Buyse et al., 2000*).

Failure to conduct a systematic review of preclinical studies prior to the start of the clinical research and presenting selective results to grant funders or patients is scientifically questionable, likely to be unethical, and can lead to delays in finding suitable treatments for diseases by investing resources in treatments that could have been predicted to fail (*Cohen, 2018*; *Ritskes-Hoitinga & Wever, 2018*). Therefore, this signalling question assesses whether the authors provide evidence from systematic reviews of preclinical animal research studies and clinical studies that the intervention or a similar intervention showed treatment effects that were similar in preclinical research studies and clinical studies in humans.

## Domain 2: experimental design and data analysis plan

The purpose of this domain is to assess the experimental study design and assess the analysis performed by the authors with respect to random errors and measurement errors. There are very good online resources that can help with the experimental design and statistical analysis in preclinical studies (*Bate & Clark, 2014*; *Festing, 2016*; *Nature Collection, 2018*). These resources can help in the assessment of this domain.

### 2.1 Did the authors describe sample size calculations?

Sample size calculations are performed to control for random errors (i.e. ensure that a difference of interest can be observed) and should be used in preclinical studies that involve hypothesis testing (for example a study conducted to find out whether a treatment is likely to result in benefit). This signalling question assesses whether the authors have described the sample size calculations to justify the number of animals used to reliably answer the research question.

### 2.2 Did the authors plan and perform statistical tests taking the type of data, the distribution of data, and the number of groups into account?

The statistical tests that are performed depend upon the type of data (for example categorical nominal data, ordinal data, continuous quantitative data, continuous discrete data), distribution of data (for example normal distribution, binomial distribution, Poisson distribution, etc.), and the number of groups compared. The authors should justify the use of statistical tests based on the above factors. The hypothesis testing should be pre-planned. This signalling question assesses whether the authors planned and performed statistical tests taking type of data, distribution of data, and the number of groups compared into account.

The authors may use multivariable analysis (analysis involving more than one predictor variable) or multivariate analysis (analysis involving more than one outcome variable), although these terms are often used interchangeably (*Hidalgo & Goodman, 2013*). Some assumptions about the data are made when multivariable analysis and multivariate analysis are performed (*Casson & Farmer, 2014*; *Nørskov et al., 2020*) and the results are reliable only when these assumptions are met. Therefore, assessment of whether the authors have reported about the assumptions should be considered as a part of this signalling question.

The authors may have also performed unplanned hypothesis testing after the data becomes available, which is a form of 'data dredging' and can be assessed in the next signalling question. The authors may also have made other changes to the statistical plan. This aspect can be assessed as part of signalling question 8.2.

### 2.3 Did the authors make adjustment for multiple hypothesis testing?

This signalling question assesses whether study authors have made statistical plans to account for multiple testing.

When multiple hypotheses are tested in the same research, statistical adjustments are necessary to achieve the planned alpha and beta errors. Testing for more than two groups is a form of multiple testing: the statistical output usually adjusts for more than two

groups. However, testing many outcomes is not usually adjusted in the statistical software output and has to be adjusted manually (or electronically) using some form of correction. This is not necessary when the study authors have a single primary outcome and base their conclusions on the observations on the single primary outcome. However, when multiple primary outcomes are used, adjustments for multiple hypothesis testing should be considered (*Streiner, 2015*). For example if the effectiveness of a drug against cancer is tested by apoptosis, cell proliferation, and metastatic potential, authors should consider statistical adjustments for multiple testing.

Multiple analyses of the data with the aim of stopping the study once statistical significance is reached and data dredging (multiple unplanned subgroup analyses to identify an analysis that is statistically significant; other names include '*P* value fiddling' or 'P-hacking') are other forms of multiple testing and should be avoided (*Streiner, 2015*). Methods for interim analysis to guide stopping of clinical trials such as sequential and group sequential boundaries have been developed (*Grant et al., 2005*). Implementation of group sequential designs may improve the efficiency of animal research (*Neumann et al., 2017*).

### 2.4 If a dose-response analysis was conducted, did the authors describe the results?

In pharmacological testing in animals, it is usually possible to test multiple doses of a drug. This may also apply to some non-pharmacological interventions, where one can test the intervention at multiple frequencies or duration (for example exercise for 20 min vs exercise for 10 min vs no exercise). A dose-response relationship indicates that the effect observed is greater with an increase in the dose. Animal studies incorporating dose-response gradients were more likely to be replicable to humans (*Hackam & Redelmeier, 2006*). This signalling question assesses whether the authors have reported the dose-response analysis if it was conducted.

### 2.5 Did the authors assess and report accuracy?

Accuracy is the nearness of the observed value (using the method described) to the true value. Depending upon the type of outcome, these can be assessed by Kappa statistics, Bland–Altman method, correlation coefficient, concordance correlation coefficient, standard deviation, or relative standard deviation (*Bland & Altman, 1986*, *1996a*, *1996b*, *1996c*; *Van Stralen et al., 2008*; *Watson & Petrie, 2010*; *Zaki et al., 2012*). This signalling question assesses whether the authors have provided a measure of accuracy by using an equipment for which accuracy information is available, or used a reference material (material with known values measured by an accurate equipment) to assess accuracy.

### 2.6 Did the authors assess and report precision?

Precision, in the context of measurement error, is the nearness of values when repeated measurements are made in the same sample (technical replicates). The same methods used for assessing accuracy can be used for assessing precision, except that instead of using a reference material, the comparison is between the measurements made in the same sample for assessing precision. The width of confidence intervals can also provide a

measure of the precision. This signalling question assesses whether the authors have measured and reported precision.

### 2.7 Did the authors assess and report sampling error?

In some situations, errors arise because of the non-homogenous nature of the tissues or change of values over time, for example diurnal variation. The same methods used to assess accuracy can be used for assessing sampling error, except that instead of using a reference material, the comparison is between the measurements made in samples from different parts of cancer/diseased tissue (biological replicates) or samples from different times. This signalling question assesses whether the authors have measured and reported sampling error.

### 2.8 Was the measurement error low or was the measurement error adjusted in statistical analysis?

This signalling question assesses whether the measurement errors (errors in one or more of accuracy, precision, sampling error) were low or were reported as adjusted in statistical analysis. There are currently no universally agreed values at which measurement errors can be considered low. This will depend upon the context and the measure used to assess measurement error. For example if the differences between the groups is in cm and the measurement error is non-differential (i.e. the error does not depend upon the intervention) and is a fraction of a mm, then the measurement error is unlikely to cause a major difference in the conclusions. On the other hand, if the measurement error is differential (i.e. the measurement error depends upon the intervention) or large relative to the effect estimates, then this has to be estimated and adjusted during the analysis. Measurement error can be adjusted using special methods such as ANOVA repeated measurements, general linear model repeated measurements, regression calibration, moment reconstruction, or simulation extrapolation (*Vasey & Thayer, 1987*; *Carroll, 1989*; *Lin & Carroll, 1999*; *Littell, Pendergast & Natarajan, 2000*; *Freedman et al., 2004*, *2008*).

## Domain 3: Bias (internal validity)

Even if an animal model with good construct validity is chosen, biases such as selection bias, confounding bias, performance bias, detection bias, and attrition bias can decrease the value of the study (*Higgins & Green, 2011*). The purpose of this domain is to assess the risks of bias in the study.

### 3.1 Did the authors minimise the risks of bias such as selection bias, confounding bias, performance bias, detection bias, attrition bias, and selective outcome reporting bias?

Some sources, examples, and rationale for the risk of bias in animal studies are available in the SYRCLE's risk of bias assessment tool for animal research, National Research Council's guidance of description of animal research in scientific publications, US National Institute of Neurological Disorders and Stroke's call for transparent reporting, and National Institute of Health's principles and guidelines for Reporting Preclinical Research (*National Research Council, 2011*; *Landis et al., 2012*; *Hooijmans et al., 2014*; *National*

*Institutes of Health, 2014*). These risks of bias should have been minimised in the study. While many researchers are familiar with most of these types of bias, selective outcome reporting warrants further discussion. Selective outcome reporting is a form of bias where study authors selectively report the results that favour the intervention. The selective outcome reporting bias should, as a minimum, cover whether the choice of results to be reported (in tables, text, or figures) were predetermined. Changing the outcomes is prevalent in human clinical trials (*Jones et al., 2015*; *Altman, Moher & Schulz, 2017*; *Howard et al., 2017*). There are no studies that investigate the prevalence of changing the outcomes in preclinical animal research; however, one can expect that it is at least as prevalent in preclinical animal research as in clinical research. It is now also possible to register preclinical animal studies at www.preclinicaltrials.eu and www.osf.io before they start, which can help with the assessment of selective outcome reporting bias.

In some situations, International Organization for Standardization (ISO) standards (*Chen & Wang, 2018*) and National Toxicology Program recommendations (*U.S. Department of Health and Human Services, 2018*) may also be applicable.

## Domain 4: reproducibility of results in a range of clinically relevant conditions (external validity)

The purpose of this domain is to assess whether the results were reproduced in a range of clinically relevant conditions (different methods of disease induction, different genetic composition, different ages, sex, etc).

### 4.1 Were the results reproduced with alternative preclinical models of the disease/condition being investigated?

The underlying rationale behind preclinical animal research is the genetic, anatomical, physiological, and biochemical similarities (one or more of the above) between animals and humans. Different animals have different levels of genetic similarities with humans and between each other (*Gibbs et al., 2004*; *Church et al., 2009*; *Howe et al., 2013*), which leads to anatomical, physiological, and biochemical differences between the different species. This can lead to differences in the treatment effects between different animal species or different models of induction of disease. The differences may be in the direction (for example the intervention is beneficial is some species and harmful in others) or in the magnitude (for example the intervention is beneficial in all the species, but the treatment effects differ in differ species). Even if the inconsistency is only in the magnitude of effect, this indicates that the treatment effects in humans may also be different from those observed in different species. Therefore, consistent treatment effects observed across different animal species or different models of induction of disease may increase the likelihood of similar treatment effects being observed in humans. This signalling question assesses the consistency across different preclinical models.

### 4.2 Were the results consistent across a range of clinically relevant variations in the model?

In the clinical setting, a treatment is used in people of different ages, sex, genetic composition, and with associated comorbidities. These differences within species and

existing comorbidities can lead to different treatment effects even if the same species and the model of induction is used. Therefore, this signalling question assesses whether animals of multiple ages, sex, genetic compositions, and existing comorbidities were used and whether the treatment effect was consistent across these clinically relevant variations.

### 4.3 Did the authors report take existing evidence into account when choosing the comparators?

Researchers may choose an inactive control rather than an established active treatment as the control to show that a drug is effective. They may also choose a weak control such as a dose lower than the effective dose or an inappropriate route for control to demonstrate a benefit of the intervention. Therefore, in these last examples, experimental results that the intervention is better than control are applicable only for the comparison of the intervention with a weak control, which may not be clinically relevant. This signalling question assesses whether the authors chose an established active treatment at the correct dose or route (in the case of pharmacological interventions) as control.

## Domain 5: reproducibility and replicability of methods and results in the same model

In a survey of more than 1,500 scientists conducted by Nature, more than 70% of researchers tried and failed to reproduce another scientist's experiments, and more than half failed to reproduce their own experiments (*Baker, 2016*). About 90% of scientists surveyed thought that there was a slight or significant 'reproducibility crisis' (*Baker, 2016*). This domain assesses the reproducibility (the ability to achieve similar or nearly identical results using comparable materials and methodologies) and replicability (the ability to repeat a prior result using the same source materials and methodologies) (*FASEB, 2016*) of the methods and results in the same animal model and differs from external validity, which focusses on whether the results were reproduced in a different clinically relevant model.

### 5.1 Did the authors describe the experimental protocols sufficiently to allow their replication?

One of the methods of improving replication is to describe the experimental protocols sufficiently. This signalling question assesses whether the authors have described the experimental protocols sufficiently to allow their replication.

### 5.2 Did an independent group of researchers replicate the experimental protocols?

This signalling question is different from the "Did the Authors Describe the Experimental Protocols Sufficiently to Allow their Replication?", above. The previous question assesses whether the protocols were described sufficiently, while this signalling question assesses whether these protocols were actually replicated by an independent group of researchers. The independent group of researchers could be part of the author team and could be from the same or different institutions, as long as they repeated the experiments

independently. The results of replication of experimental protocols can be part of the same report, but could also be another report.

### 5.3 Did the authors or an independent group of researchers reproduce the results in similar and different laboratory conditions?

This signalling question is different from the "Did the Authors Describe the Experimental Protocols Sufficiently to Allow their Replication?", above. The previous question assesses whether the protocols were protocols could be replicated. This signalling questions assesses whether the results could be reproduced in similar and different laboratory conditions. Even when the protocols/methods are replicated by an independent group of researchers, the results may not be replicated or reproduced in similar and/or different laboratory conditions (*Baker, 2016*). This signalling question assesses whether the results were replicated or reproduced. Attempts to replicate or reproduce the results can be a part of the same report, but could also be another report, particularly if the attempt to replicate or reproduce the results is made by an independent group of researchers.

## Domain 6: implications of the study findings (study conclusions)

The purpose of the domain is to assess whether the authors have made conclusions that reflect the study design and results.

### 6.1 Did the authors' conclusions represent the study findings, taking its limitations into account?

This signalling question assesses whether the study authors considered all the findings and limitations of the study while arriving at conclusions. The study authors may have made conclusions based on extrapolations of their results and not on their data, which is poor research practice. This should also be considered while assessing this signalling question. Studies designed to look at pathophysiology of disease or mechanism of action of treatment should demonstrate evidence of similarity between disease process in animal and human disease before arriving at conclusions regarding these aspects.

### 6.2 Did the authors provide details on additional research required to conduct first-in-human studies?

Researchers should consider the limitations of their study before recommending first-in-human studies. For example this may be the first experimental study on this research question; therefore, the research question may not have been conducted in multiple centres. The authors should highlight the need for studies that reproduce the results by a different group of researchers. If the current study was a study to attempt reproduction of the results of a previous study, then the authors should clarify whether further preclinical studies are required or whether the intervention should be evaluated in humans with justifications: repeating the study in preclinical models can be justified if the intervention needs to be evaluated after a modification; recommending evaluation in humans can be justified if efficacy and safety has been demonstrated consistently in multiple preclinical models and centres.

This signalling question assesses whether the study authors have made future research recommendations based on the study design and results from this study in the context of other studies on this issue. A study on investigator brochures in Germany demonstrated that animal study results were not evaluated well, for example by systematic reviews of animal studies before clinical trials (*Wieschowski et al., 2018*), highlighting that the further research recommendations should be made taking other studies on the topic into account.

## Domain 7: research integrity

The purpose of this domain is to ensure that the authors adhered to the principles of research integrity during the design, conduct and reporting of their research. If the authors did not adhere to the principles of research integrity, the results can be unreliable even if the study experimental design and analysis were reliable. Lack of research integrity can decrease the confidence in the study findings.

### 7.1 Did the research team obtain ethical approvals and any other regulatory approvals required to perform the research prior to the start of the study?

Animal research should be performed ethically in a humane way. While university ethics boards can confirm the existence of ethical approval, additional licencing requirements (for example Home Office License in UK) may be necessary before the research can be conducted. This is to ensure that the principles of replacement (methods which avoid or replace the use of animals), reduction (methods which minimise the number of animals used per experiment), and refinement (methods which minimise animal suffering and improve welfare) are followed during scientific research (*NC3Rs*; *UK Government, 1986*). In some countries like the UK, preclinical studies conducted to justify human clinical trials are required to follow Good Laboratory Practice Regulations (*UK Government, 1999*). This signalling question assesses whether the study authors have provided the details of ethics approval or any other regulatory approvals and standards that they used in their research.

### 7.2 Did the authors take steps to prevent unintentional changes to data?

Unintentional human errors when handling data ('data corruption') has the potential to affect the quality of study results and a possible reason for lack of reproducibility as they can cause misclassification of exposure or outcomes (*Van den Broeck et al., 2005*; *Ward, Self & Froehle, 2015*). 'Data cleaning' is the process of identifying and correcting these errors, or at least attempting to minimise the impact on study results (*Van den Broeck et al., 2005*). Methods used for data cleaning can have a significant impact on the results (*Dasu & Loh, 2012*; *Randall et al., 2013*). The best way to minimise data errors is to avoid them in the first place. While there are many 'data handling' guidelines about the protection of personal data, there is currently no guidance on the best method to avoid 'data corruption'. The UK Digital Curation Centre (www.dcc.ac.uk) provides expert advice and practical help to research organisations wanting to store, manage, protect, and share digital research data. Maintenance of laboratory logs, accuracy measures between laboratory logs and data used, and use of password-protected data files can all decrease the

risks of unintentional changes to data. This signalling question assesses whether the authors took steps to prevent unintentional changes to data.

## Domain 8: research transparency

The purpose of this domain is to assess whether the animal study authors were transparent in their reporting. Transparent reporting increases the confidence in the study findings and promotes replicability of the research findings. Reporting guidelines such as ARRIVE guidelines 2.0, Gold Standard Publication Checklist to improve the quality of animal studies, and National Research Council's guidance on description of animal research in scientific publications can help with transparent reporting (*Hooijmans, Leenaars & Ritskes-Hoitinga, 2010*; *National Research Council, 2011*; *Percie du Sert et al., 2020*).

### 8.1 Did the authors describe the experimental procedures sufficiently in a protocol that was registered prior to the start of the research?

While selective outcome reporting is covered under the bias (internal validity) domain, the authors may have changed the protocol of the study in various other ways, for example the disease-specific model, intervention, control, or the methods of administration of the intervention and control. The experimental protocols should be registered prior to the start of the study in a preclinical trial registry, such as, https://www.preclinicaltrials.eu/, which allows registration of animal studies and is searchable. Studies can also be registered in Open Science Framework (https://osf.io/). Alternatively, posting the protocol in open access preprint servers such as https://www.biorxiv.org/ or https://arxiv.org/, print or online journals, in an institutional or public data repository such as https://zenodo.org/ is another option. The study authors should provide a link to this registered protocol in their study report.

The focus of this signalling question is about availability of a registered protocol prior to research commencement, which had enough details to allow replication, while the signalling question "Did the Authors Describe the Experimental Protocols Sufficiently to Allow their Replication?" refers to the description of the final protocol used (after all the modifications to the registered protocol) in sufficient detail to allow replication.

### 8.2 Did the authors describe any deviations from the registered protocol?

There may be justifiable reasons for alteration from a registered protocol. The authors should be explicit and describe any deviations from their plans and the reasons for them. In addition to registries, repositories, and journals for registering preclinical trials, some journals also offer 'registered reports' publishing format, which involves peer review of the study design and methodology, and if successful, results in a conditional acceptance for publication prior to the research being undertaken (*Hardwicke & Ioannidis, 2018*). This will also allow evaluation of the deviations from the registered protocol.

### 8.3 Did the authors provide the individual subject data along with explanation for any numerical codes/substitutions or abbreviations used in the data to allow other groups of researchers to analyse?

In addition to making the protocol available, the key aspects of reproducibility and replicability in research involving data are the availability of the raw data from which results were generated, the computer code that generated the findings, and any additional information needed such as workflows and input parameters (*Stodden, Seiler & Ma, 2018*). Despite the journal policies about data sharing, only a third of computational and data analysis could be reproduced in a straightforward way or with minor difficulty (*Stodden, Seiler & Ma, 2018*). The remaining required substantial revisions for reproduction or could not be reproduced (*Stodden, Seiler & Ma, 2018*).

During the analysis, the authors may have processed the data to allow analysis. This may be in the form of transformation of data (for example log-transformation or transformation from continuous or ordinal data into binary data), substitutions of texts with numbers (for example intervention may be coded as 1 and control may be coded as 0; similarly, the characteristics and/or outcomes may have been coded), or may have used abbreviations for variable names to allow easy management and meet the requirements for the statistical software package used. Some authors may use complex computer codes to perform the analysis. This is different from the transformation or substitution codes and refers to a set of computer commands that are executed sequentially by the computer. While the authors may provide the individual subject data as part of data sharing plan or as a journal requirement, this data is unlikely to be useful for analysis if the transformation codes, substitution codes, abbreviations, or computer codes are not available. Therefore, the individual participant data should be provided along with any transformation codes, substitution codes, and abbreviations to allow other researchers to perform analysis. The individual participant data can be provided either as a supplementary appendix in the journal publication or can be provided in open access repositories such as https://zenodo.org/ or university open access repositories. This signalling question assesses whether individual subject data with sufficient details to reanalyse were available.

## DISCUSSION

Using a modified Delphi consensus process, we have developed a tool to assess the clinical relevance of a preclinical research study in terms of the likelihood that therapeutic preclinical research methods and findings can be translated into improvement in the management of human diseases. We searched for existing guidelines about the design, conduct, and reporting of preclinical research and developed domains and signalling questions by involving experts. A modified Delphi consensus process was used to develop new domains and signalling questions and refine the existing domains and signalling questions to improve the understanding of the people who assess the clinical relevance of animal research. We have included only questions for which consensus was achieved (i.e. at least 70% of the Delphi panel members considered the question important to evaluate the clinical relevance of animal research). This tool provides a

framework for researchers, journal editors, grant funders, and regulatory authorities to assess the clinical relevance of preclinical animal research with the aim to achieve better design, conduct, and reporting of preclinical animal research.

This tool is different from the ARRIVE guidelines 2.0 (*Percie du Sert et al., 2020*) and the NIH effort on improving preclinical research (*National Institutes of Health, 2014*) as our tool is a domain-based assessment tool rather than a reporting guideline. Furthermore, as opposed to a reporting guideline where the questions relate to clarity of reporting, the questions in this tool assess the likelihood of the results being clinically relevant. This tool is also different from the SYRCLE risk of bias of tool, as this tool goes beyond the risk of bias in the research (*Hooijmans et al., 2014*). While many of the issues have been covered by other reporting guidance on preclinical research, the issue of measurement errors (errors in accuracy, precision, or sampling error) have not been addressed in existing guidance on preclinical research. Measurement error in exposure or outcome is often neglected in medical research despite the potential to cause biased estimation of the effect of an exposure or intervention (*Hernan & Cole, 2009*; *Brakenhoff et al., 2018a*, *2018b*). Even though preclinical animal research often involves repeated measurements, the measurement error is generally not reported or not taken into account during the analysis. This Delphi panel arrived at a consensus that measurement errors should be taken into account during the analysis if necessary and should be reported to enable an assessment of whether the preclinical research is translatable to humans.

We are now piloting this tool to improve it. This is in the form of providing learning material to people willing to pilot this tool and requesting them to assess the clinical relevance of preclinical animal studies. Financial incentives are being offered for piloting the tool. We intend to pilot the tool with 50 individuals including researchers performing or planning to perform preclinical or clinical studies. If the percentage agreement for classification of a domain is less than 70%, we will consider refining the question, explanation, or training by an iterative process to improve the agreement. The link for the learning material is available at: https://doi.org/10.5281/zenodo.4159278. The tool can be completed using an Excel file, which is available in the same link.

## CONCLUSIONS

We have developed a tool to assess the clinical relevance of preclinical studies. This tool is currently being piloted.

### Funding

The authors received no funding for this work.

### Competing Interests

Nicola Osborne is the founding member of Responsible Research in Practice.
The authors declare that they have no competing interests.

## Author Contributions

- Kurinchi S. Gurusamy conceived and designed the experiments, performed the experiments, analysed the data, prepared figures and/or tables, authored or reviewed drafts of the paper, and approved the final draft.
- David Moher conceived and designed the experiments, performed the experiments, authored or reviewed drafts of the paper, and approved the final draft.
- Marilena Loizidou conceived and designed the experiments, performed the experiments, authored or reviewed drafts of the paper, and approved the final draft.
- Irfan Ahmed performed the experiments, authored or reviewed drafts of the paper, and approved the final draft.
- Marc T. Avey performed the experiments, authored or reviewed drafts of the paper, and approved the final draft.
- Carly C. Barron performed the experiments, authored or reviewed drafts of the paper, and approved the final draft.
- Brian Davidson performed the experiments, authored or reviewed drafts of the paper, and approved the final draft.
- Miriam Dwek performed the experiments, authored or reviewed drafts of the paper, and approved the final draft.
- Christian Gluud conceived and designed the experiments, performed the experiments, authored or reviewed drafts of the paper, and approved the final draft.
- Gavin Jell performed the experiments, authored or reviewed drafts of the paper, and approved the final draft.
- Kiran Katakam performed the experiments, authored or reviewed drafts of the paper, and approved the final draft.
- Joshua Montroy performed the experiments, authored or reviewed drafts of the paper, and approved the final draft.
- Timothy D. McHugh performed the experiments, authored or reviewed drafts of the paper, and approved the final draft.
- Nicola J. Osborne performed the experiments, authored or reviewed drafts of the paper, and approved the final draft.
- Merel Ritskes-Hoitinga performed the experiments, authored or reviewed drafts of the paper, and approved the final draft.
- Kees van Laarhoven performed the experiments, authored or reviewed drafts of the paper, and approved the final draft.
- Jan Vollert performed the experiments, authored or reviewed drafts of the paper, and approved the final draft.
- Manoj Lalu conceived and designed the experiments, performed the experiments, authored or reviewed drafts of the paper, and approved the final draft.

## Data Availability

The data are available as a Supplemental File.

## Supplemental Information

Supplemental information for this article can be found online at http://dx.doi.org/10.7717/peerj.10673#supplemental-information.

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
