# Peer review of "Clinical relevance assessment of animal preclinical research (RAA) tool: development and explanation"

_PeerJ, doi:10.7717/peerj.10673_

## Round 0.1 · original submission · Major Revisions

Your manuscript was considered interesting and valuable by both reviewers. Both reviewers commended you on the tremendous amount of work you did to develop this much needed tool. However the reviewers raised a number of points that need to be addressed and that they believe will improve your manuscript.

Please, submit a detailed rebuttal which shows where and how you have taken all comments and suggestions into consideration. If you do not agree with some of the reviewers’ comments or suggestions, please explain why. Your rebuttal will be critical in making a final decision on your manuscript. Please, note also that your revised version may enter a new round of review by the same or by different reviewers. Therefore, I cannot guarantee that your manuscript will eventually be accepted.

·

Basic reporting

See below

Experimental design

See below

Validity of the findings

See below

Additional comments

The authors of this study used a modified Delphi approach to develop a tool to assess the clinical relevance of a preclinical research study and assess the likelihood that the therapeutic preclinical research methods and findings would be successfully translated into improvement in the management of human diseases. The process identified eight domains and 28 signaling questions that can provide a framework for researchers, journal editors, grant funders, and regulatory authorities to assess the clinical relevance of preclinical animal research, going beyond reporting standards and minimization of bias in study design.

The authors should be applauded for tackling this challenging yet critical topic. They have done a tremendous amount of work and their product will be helpful for a long list of stakeholders. Please see below for issues/clarifications to address:

1. Lines 173-205: A flow diagram specifically outlining how many questions were included or excluded at each stage would be helpful to clarify the specifics of each Delphi step.
2. Please clarify: Were all steps completed electronically? What platform was used?
3. Lines 221-226: Please provide a table summarizing the demographics of the Delphi panel members.
4. Did the panel members declare conflicts of interest?
5. There are distracting font changes throughout (example line 420).
6. Lines 429-438: This paragraph is very difficult to interpret. Please rewrite.
7. Line 630: Please clarify, repeated/replicated within the same study?
8. Please clarify the difference between signaling questions 5.2 and 5.3.
9. Lines 792-797: As a major point, I request that the authors provide more detail on the piloting process, as well as practical guidance/direction on how the framework can be used for an individual study. How many positive responses would be considered an overall positive assessment? Please complete the manuscript with clearer future steps for the reader.

·

Basic reporting

The authors present development and explanation of a tool to assess the clinical relevance of preclinical research. Such a tool is very much welcomed as only a small proportion of preclinical research translates into clinical benefit in humans. The proposed tool assesses the likelihood that therapeutic preclinical findings can be translated into improvement of the management of human diseases. The intended users of the tool are researchers conducting first-in-human studies, journal editors, grant funders, and regulatory authorities. The tool is currently being piloted.

The manuscript is long, but reads well. Overall the reporting is clear and unambiguous and professional English language is used throughout the manuscript. The structure of the manuscript is clear and the tables and figures are relevant.

I do have some comments on the structure of the Introduction. The second paragraph of the Introduction (line 97-119) is a description of the Methods and Results and should not be part of the Introduction. What I miss in the Introduction is a thorough analysis and discussion of the reasons and explanations for the lack of translation of the results observed in preclinical research, and a framework or model of how translatability works, or might work. This is mentioned in just one sentence; in my opinion, this should be the backbone of the tool.

Line 120: In ‘Why is this project needed?’ the authors explain the choice for the type of tool (domain-based). In this paragraph I would like to read something about other tools with the same goal, if any, to put development of the proposed RAA tool into context. Please consider moving the choice for the type of tool move to the Methods section.

Line 134: Please explain what first-in-human studies are. This means that not all preclinical research is relevant for this tool.

Experimental design

In general, the method used to develop a tool like this (literature search plus Delphi), and the design of the tool (domain-based) are both well established and state of the art.

I have however some comments and questions on the approach.

The authors followed the Guidance for Developers of Health Research Reporting Guidelines as general method. Could you please explain why you choose reporting as a starting point? The lack of translatability is about much more than adequate reporting. We know of course that reporting guidelines are often used to assist in designing studies, but guidance to develop a methodological (or conceptual or analytical) framework as starting point would have seemed more logical and appropriate to me. A related point is the search (line 141), please explain why you choose a snowballing technique over a systematic search.

Line 148-149: The topics covered in the previous guidance were recorded. Can the authors please explain how they decided which items from the guidance papers should be recorded? For example, how did they deal with duplicate items or topics? Please provide this list as supplemental information.

Line 153: In Box 1, point 2 is not about signaling questions but domains, and addresses another point, namely how to use the tool. Looking at what is said here, the authors state that certain domains may be more important than the other, depending upon the nature and purpose for the research. It would be very helpful for the tool user to have a list of examples of the different nature and purposes and a suggestion for which domain should receive more weight, given that specific nature or purpose.

Box 2 (line 172) presents an overview of the Delphi-method. What I miss is a connection of the items/topics/questions that are found in the literature and the reasons for, or aspects of the lack of translation of the results of preclinical research. From what the authors describe it seems that the Delphi panel members prioritized and scored all items/topics/questions that were found in the literature. What is for each domain the rationale and evidence that the domain contributes to, or is an aspect of translatability of results of preclinical studies, and should therefore be included in the tool?

Line 173: Box 2, point 1: please provide these questions as supplemental information. It is not clear of the first round only included questions regarding scope and necessity of the tool, or that these questions were early formulations of the signaling questions and domains (that seems so, based on point 3 and 4).
Line 202: Box 2, point 8: what is meant by aspects?
Did the authors already present the domain structure and its topics to the panel, or was this a result from the Delphi rounds? Please add more details of how the tool got his shape and format.

Validity of the findings

The domains and signaling questions are very comprehensive and cover a very broad range of topics. Applying all these signaling questions give in-depth insight in the preclinical study that is assessed. My concern however is that the connection with translatability is often not clear.

Line 247: What is ‘low concerns for the domain’? Concerns for what?
The tool assesses the likelihood that preclinical findings can be translated into improvement in the management of human diseases. How does likelihood relate to concerns for the domain? I would expect an outcome like ‘likely’, ‘probably likely’, ‘probably not likely’, ‘not likely’.
Line 252: I find the statement that the tool is not for assessment of the quality of the study, as the first 4 domains cover construct validity, internal and external validity and bias, all aspects of how well the study was conducted.

Line 275/6: How is clinical relevance in terms of translation defined? Does it mean that the same results can be expected in humans? Or the same direction of the effect? Please define clearly what is meant by ‘clinical relevance’.
For each domain answering the signaling questions gives a low, moderate or high concern outcome. What is the judgement across domains? The authors state that low concerns for all domains means high clinical relevance, but what is the judgment when there are low concerns for some domains and high concerns for others?
What kind of decisions can be based on the end result of the assessment? Please provide explanation and examples of how use of the tool can guide decisions for the users.

Line 284: Scoring: please add user’s guidance for how to use the tool in practice. Should the tool be used by two assessors independently, for example? What level of knowledge and expertise is needed to use the tool?
It is the preclinical study that is assessed. I assume this is done with a clinical question in mind, and this should be made explicit by both the preclinical researcher and the user of the tool. Adding preparatory steps before answering the signaling questions, like for example in the QUADAS-2 tool (flow chart of the study) and ROBINS-I tool (target trial and confounders) would enhance usability of the tool.

Line 294 etc: Rationale and explanation of domains and signaling questions.
Domain 6: Implication of the study findings (study conclusions): why this is important for assessing the clinical relevance? This assessment can be made based on the study design, the choice of the animal model(s), the comparisons etc. How the preclinical researchers draw conclusions about their findings are not relevant, in my opinion.
Domain 7: line 671: ‘If the authors did not adhere … even if the study experimental design and analysis were reliable’ – please provide a rationale and references for this statement.

Please provide detailed information about the pilot (how, and for which users), and add the learning material as supplemental files.

Additional comments

Thank you for developing this much needed tool. I read your manuscript with great interest. My main concern is that I often miss the rationale and evidence for how the domains are related to translatability and that the construction of tool is so comprehensive and complex that it will be very hard to use. I hope the pilot will give insight in this, and I think the results of the pilot will be crucial for how the tool will further develop. I find it hard to make a final judgment about the manuscript without having seen the results of the pilot. Another main concern is the scoring, for me the aim of assessing likelihood that preclinical findings translate into improvement in the management of human disease and scoring concerns with the domains is not logical.

---

## Round 0.2 · accepted · Accept

You did a comprehensive and thorough job of addressing the numerous comments provided by the reviewers, resulting in an improved manuscript.

[# Staff note - the Editor also evaluated, and was satisfied by, the responses to the comments by Reviewer 2 #]

·

Basic reporting

Appropriate for publication.

Experimental design

Appropriate for publication.

Validity of the findings

Appropriate for publication.

Additional comments

Thank-you for addressing my comments. I look forward to your next publication on the results of piloting the new tool you have developed.